# Modeling and Optimization of an Enhanced Soft Sensor for the Fermentation Process of *Pichia pastoris*

**DOI:** 10.3390/s24103017

**Published:** 2024-05-09

**Authors:** Bo Wang, Ameng Yu, Haibo Wang, Jun Liu

**Affiliations:** Key Laboratory of Agricultural Measurement and Control Technology and Equipment for Mechanical Industrial Facilities, School of Electrical and Information Engineering, Jiangsu University, Zhenjiang 212013, China; wangbo@ujs.edu.cn (B.W.); 2212107036@stmail.ujs.edu.cn (H.W.); liujun4503@126.com (J.L.)

**Keywords:** soft sensor, *Pichia pastoris*, transfer component analysis, maximal information coefficient, least squares support vector machine, improved northern goshawk optimization

## Abstract

This paper proposes a novel soft sensor modeling approach, MIC-TCA-INGO-LSSVM, to address the decline in performance of soft sensor models during the fermentation process of *Pichia pastoris*, caused by changes in working conditions. Initially, the transfer component analysis (TCA) method is utilized to minimize the differences in data distribution across various working conditions. Subsequently, a least squares support vector machine (LSSVM) model is constructed using the dataset adapted by TCA, and strategies for improving the northern goshawk optimization (INGO) algorithm are proposed to optimize the parameters of the LSSVM model. Finally, to further enhance the model’s generalization ability and prediction accuracy, considering the transfer of knowledge from multiple-source working conditions, a sub-model weighted ensemble scheme is proposed based on the maximum information coefficient (MIC) algorithm. The proposed soft sensor model is employed to predict cell and product concentrations during the fermentation process of *Pichia pastoris*. Simulation results indicate that the *RMSE* of the INGO-LSSVM model in predicting cell and product concentrations is reduced by 47.3% and 42.1%, respectively, compared to the NGO-LSSVM model. Additionally, TCA significantly enhances the model’s adaptability when working conditions change. Moreover, the soft sensor model based on TCA and the MIC-weighted ensemble method achieves a reduction of 41.6% and 31.3% in the *RMSE* for predicting cell and product concentrations, respectively, compared to the single-source condition transfer model TCA-INGO-LSSVM. These results demonstrate the high reliability and predictive performance of the proposed soft sensor method under varying working conditions.

## 1. Introduction

As one of the most widely used protein expression systems in current applications, the *Pichia pastoris* expression system has been extensively employed for expressing various exogenous proteins such as serum proteins, insulin, and epidermal growth factor, among others [1]. Among these, cell concentration and product concentration are crucial process variables in the fermentation process, and the real-time and accurate detection of their values are essential for improving production efficiency and optimizing control [2]. For the measurement of key biological parameters during the fermentation process, offline sampling and laboratory analysis are mostly adopted [3]. However, these approaches not only contaminate the fermentation environment but also fail to provide real-time results. Therefore, the real-time and accurate acquisition of key biological parameters during the fermentation process of *Pichia pastoris* has become an important research topic.

Soft sensor technology emerges as a solution to the aforementioned issues [4]. It has been widely utilized globally and extensively researched by scholars, yielding fruitful results. For instance, Lu et al. [5] proposed a weighted ensemble learning soft sensor modeling method based on an improved seagull optimization algorithm and Gaussian process regression to address the issue of traditional single methods failing to describe the nonlinear characteristics of the entire fermentation process. Applying this soft sensor method to predict key biochemical parameters in the fermentation process of marine lysozyme, the results indicate relatively small errors. Song et al. [6] proposed a virtual sample generation method based on data augmentation and weighted interpolation to expand the soft sensor dataset with high-quality samples. To verify the effect of the proposed method, simulations of the numerical function and the actual chemical processing of pure terephthalic acid were performed, and correlation analysis was introduced as a measure of whether the generated samples were consistent with the real ones. The results showed that the proposed model can boost the predictive power of the soft sensor by generating higher quality and more reasonable samples compared to other advanced methods.

However, in the fermentation process of *Pichia pastoris*, variations in working conditions arise from factors such as environmental changes and feeding practices [7]. These variations result in different distributions of fermentation process data under various working conditions, failing to meet the assumption of identical distribution between testing and modeling data required by the models discussed above. This discrepancy challenges the adaptability of traditional soft sensor models to changing working conditions. Consequently, applying a soft sensor model established under the source working condition to new working conditions results in diminished predictive performance. To address this issue, it is essential to enhance the soft sensor model to ensure that it maintains excellent predictive performance under varying working conditions.

In order to address the problem of the soft sensor model’s degraded predictive ability under the target working condition, researchers have introduced transfer learning. Transfer learning aims to improve the performance of target learners on target domains by transferring the knowledge contained in different but related source domains [8]. For instance, Chai et al. [9] introduced deep transfer learning to soft sensor modeling and proposed a deep probabilistic transfer regression framework to enhance the target soft sensor performance. The effectiveness of the proposed method was validated through an industrial multiphase flow process. Li et al. [10] proposed an adversarial transfer learning methodology to enhance soft sensor learning to acquire more training data. The effectiveness of the proposed soft sensor and the rationale analyzer was validated in a simulated wastewater plant, benchmark simulation model No.2, and a full-scale oxidation ditch wastewater plant. Wang et al. [11] proposed a soft sensor modeling method combining a long short-term memory network and balanced distribution adaptation method, which solves the problem of soft sensor modeling under unknown modes of multiple working conditions in the *Pichia pastoris* biochemical reaction process, achieving the prediction of key parameters under different working conditions. Zhang et al. [12] proposed an online transfer kernel recursive algorithm for soft sensor modeling under changing working conditions. The proposed method incorporates the concept of online transfer learning and takes into account the nonlinearity of the process. This allows for the establishment of a kernel online recursive method in the latent variable space, integrating parameter transfer and sample transfer. Experimental results conducted on multiple industrial datasets validated the effectiveness of the proposed approach. The above studies have introduced the idea of transfer learning into soft sensor methods and successfully improved the predictive performance of soft sensor models under various working conditions. As a classical transfer learning algorithm, TCA tries to learn some transfer components across domains in a reproducing kernel Hilbert space using maximum mean discrepancy [13]. In the subspace spanned by these transfer components, data properties are preserved, and data distributions in different domains are close to each other. Recently, TCA has been widely used in fault detection [14], image categorization [15], sentiment detection [16], healthcare [17], and so on. However, in practice, labeled data may be collected from multiple sources, while naive application of the single-source TL algorithms may lead to suboptimal solutions [18].

Based on the above analysis, this paper proposes a soft sensor modeling method based on TCA and the MIC-weighted ensemble method to predict the cell concentration and product concentration in the fermentation process of *Pichia pastoris* under the target working condition. First, considering the inconsistency in the distribution of data between source and target working conditions of the fermentation process of *Pichia pastoris*, TCA is employed to reduce the difference in the distribution of data between the source and target fermentation processes, and the LSSVM optimized with the INGO algorithm is utilized as the regression modeling algorithm. Secondly, labeled data originate from multiple working conditions; thus, knowledge transfer solely from a single-source condition cannot achieve the desired predictive performance. As a result, this paper establishes sub-models on multiple-source condition datasets and proposes a novel weighting allocation method based on the MIC to assign appropriate weights to each sub-model, facilitating a weighted ensemble. The proposed method is applied to predict the cell concentration and product concentration under the target working condition, and the simulation results demonstrate that the INGO algorithm, compared to the NGO algorithm, is capable of seeking superior parameters for the LSSVM, thereby effectively enhancing the prediction accuracy. Furthermore, the transfer soft sensor modeling method based on the MIC-weighted ensemble proves suitable for the fermentation process of *Pichia pastoris* under various working conditions. Compared to existing soft sensor models, the proposed method exhibits superior generalization ability and predictive performance, enabling precise real-time monitoring of cell concentration and product concentration under different working conditions.

The following is the chapter organization of the paper:Introduction: This section introduces the research background and significance of the paper. It analyzes the disadvantages of existing soft sensor models when working conditions change and proposes a new soft sensor model to address these shortcomings.Materials and Methods: this section describes the relevant algorithms of the soft sensor model and explains the reasons for using these algorithms in the context of the fermentation process of *Pichia pastoris*.Results: this section applies the proposed soft sensor model to predict cell concentration and product concentration in the fermentation process of *Pichia pastoris* and compares and analyzes the simulation results.Discussion: this section discusses the results, explores their impact on the industrial application of the fermentation process of *Pichia pastoris*, identifies limitations of the current model, and outlines potential future research directions.Conclusions: this section provides a summary of the research content and conclusions of the paper.

## 2. Materials and Methods

### 2.1. Transfer Component Analysis

Traditional machine learning is characterized by training data and testing data with the same input feature space and the same data distribution; however, when there is a difference in data distribution between the training data and testing data, the results of a predictive learner can be degraded [19]. In the case of the fermentation process data under different working conditions of *Pichia pastoris*, distribution variances arise due to variations in fermentation environments. Consequently, the soft sensor model established under the source working condition demonstrates unsatisfactory predictive capability when applied to the target working condition. Transfer component analysis, by integrating cross-domain knowledge transfer strategies, effectively addresses the generalization shortcomings of traditional methods in new domains. This technique enhances the model performance in the target domain by precisely analyzing and aligning components between the source and target domains, thereby improving the predictive accuracy and adaptability of the model [20]. Therefore, this paper will employ transfer component analysis to enhance the predictive performance of the soft sensor model under the target working condition.

In 2011, Pan et al. proposed transfer component analysis (TCA), which is a classic method based on adaptive data distribution [21]. Primarily, it achieves transfer learning by minimizing the disparity in marginal probability distributions between the source and target domains. The main principles of TCA are as follows.

Given a source domain $Q_{s}={\left\{x_{s_{i}},y_{s_{i}}\right\}}_{i=1}^{n}$ and a target domain $Q_{t}={\left\{x_{t_{j}},y_{t_{j}}\right\}}_{j=1}^{m}$, suppose the feature space $X_{s}=X_{t}$ and label space $Q_{t}={\left\{x_{t_{j}},y_{t_{j}}\right\}}_{j=1}^{m}$ are the same for both domains, but the marginal probability distributions and conditional probability distributions are different, i.e., $P\left(x_{s}\right)\neq P\left(x_{t}\right)$; $P\left(y_{s}|x_{s}\right)\neq P\left(y_{t}|x_{t}\right)$. The core idea of TCA is to assume the existence of a feature mapping matrix ϕ, where the mapped data can achieve $P\left(\phi \left(x_{s}\right)\right)\approx P\left(\phi \left(x_{t}\right)\right)$. When the marginal probability distributions of data in both domains are similar, it is assumed that the conditional probability distributions are also similar, i.e., $P\left(y_{s}|\phi \left(x_{s}\right)\right)\approx P\left(y_{t}|\phi \left(x_{t}\right)\right)$. Therefore, the goal of TCA is to find a suitable ϕ.

Essentially, minimizing the difference in marginal probability distributions between the source and target domains involves finding a transformation that diminishes the distance between the two domains. TCA leverages the maximum mean discrepancy (MMD) to approximate the dissimilarity between the source and target domains. Assuming there are *n* samples in the source domain and *m* samples in the target domain, the MMD distance between these two domains can be expressed as:(1)$$Dis\left(Q_{s},Q_{t}\right)=\frac{1}{n}\sum_{i=1}^{n}\phi \left(x_{s_{i}}\right)-\frac{1}{m}\sum_{j=1}^{m}\phi {\left(x_{t_{j}}\right)}_{H}^{2}$$

To solve for ϕ, TCA introduces the kernel matrix K and the MMD matrix L, transforming the distance into the following form:(2)tr(KL)−δtr(K)
where tr(⋅) denotes the trace of the matrix, and the elements of matrix L are calculated as follows:(3)$${\left(L\right)}_{i,j}=\left\{\begin{array}{c} \frac{1}{n^{2}},x_{i},x_{j}\in Q_{s}\\ \frac{1}{m^{2}},x_{i},x_{j}\in Q_{t}\\ \frac{-1}{nm},\mathrm{otherwise} \end{array}\right.$$

At this point, the authors of TCA propose constructing the result using a matrix W with a lower dimensionality than the kernel matrix K:(4)$$\tilde{K}=\left(KK^{-\frac{1}{2}}\tilde{W}\right)\left({\tilde{W}}^{T}K^{-\frac{1}{2}}K\right)=KWW^{T}K$$

Finally, the optimization objective of TCA is:(5)$$\begin{array}{c} \underset{W}{\min}tr\left(W^{T}KLKW\right)+\lambda tr\left(W^{T}W\right)\\ \;s.t.W^{T}KHKW=I_{m} \end{array}$$
where H is the centering matrix, and $H=I_{n+m}-\frac{1}{n+m}11^{T}$.

When given two feature matrices as inputs, the first step is to compute matrices L and  H. Subsequently, utilizing the kernel function for mapping, matrix  K is calculated, and finally, W is derived to obtain the solution to the problem.

### 2.2. Improved Northern Goshawk–Least Squares Support Vector Machine Algorithm

#### 2.2.1. Least Squares Support Vector Machine

To achieve real-time and accurate monitoring of key biological parameters, such as cell concentration and product concentration, in the fermentation process of *Pichia pastoris*, it is imperative to establish a regression model between auxiliary variables and main variables. Given the nonlinear characteristics in the fermentation process of *Pichia pastoris*, the least squares support vector machine (LSSVM) is deemed suitable for the nonlinear, small sample datasets. Consequently, this paper adopts the LSSVM as the regression modeling algorithm.

For the given dataset $T=\left\{\left(x_{1},y_{1}\right),\left(x_{2},y_{2}\right),\cdots ,\left(x_{n},y_{n}\right)\right\}$, the regression function can be defined as $f(x)=w^{T}+b$, where *x* represents the sample inputs and *y* represents the sample outputs. Here, *w* and *b* are, respectively, the normal vector and the intercept of the hyperplane in a high-dimensional space. According to the principle of risk minimization, the regression problem can be transformed into a constrained optimization problem:(6)$$\left\{\begin{array}{l} \underset{\omega ,e}{\min}J(\omega ,e)=\frac{1}{2}\omega^{T}\omega +\frac{1}{2}\gamma \sum_{i=1}^{N}e_{i}^{2}\\ \;s.t.\;y_{i}=\omega^{T}\varphi \left(x_{i}\right)+b+e_{i},\;i=1,2,\cdots ,N \end{array}\right.$$
where $e_{i}$ represents the slack variable and *γ* denotes the regularization factor. By introducing Lagrange multipliers *α*:(7)$$L(\omega ,b,e,\alpha )=J(\omega ,e)-\sum_{i=1}^{N}\alpha_{i}\omega^{T}\varphi \left(x_{i}\right)+b+e_{i}-y_{i}$$

Through partial differentiation with respect to *w*, *b*, *e*, *α* the optimal values can be obtained, and the regression function established as follows:(8)$$y(x)=\sum_{i=1}^{N}\alpha_{i}K\left(x,x_{i}\right)+b$$
where $K\left(x_{i},x\right)$ is the kernel function, which comes in various types such as radial basis function (RBF) and polynomial function. In this paper, the RBF is employed as the kernel function:(9)$$K\left(x_{i},x\right)=\exp\left(-\frac{1}{2\sigma^{2}}x-{x_{i}}^{2}\right)$$

In the equation, there is only one parameter σ to be determined, representing the width of the radial basis function. According to the LSSVM regression theory, the LSSVM involves an adjustable parameter γ [22]. Therefore, when applying the LSSVM method with the RBF kernel function, there are two adjustable parameters. However, the traditional parameter selection methods are typically based on experience and experimentation, which may not guarantee the accuracy and computational efficiency of the regression model. Therefore, in order to enhance the prediction accuracy of the LSSVM, this paper proposes an improvement to the NGO algorithm for optimizing the parameters of LSSVM.

#### 2.2.2. Improved Northern Goshawk Optimization Algorithm

The northern goshawk optimization (NGO) algorithm was proposed by Mohammad Dehghani et al. in 2021 [23]. This algorithm simulates the hunting process of the northern goshawk and consists of two phases: prey identification (global search) and chase and escape operation (local search).

(1)Phase 1: Prey Identification (Exploration)

The northern goshawk, in the first phase of hunting, randomly selects a prey and then quickly attacks it. This phase increases the exploration power of the NGO due to the random selection of prey in the search space. This phase leads to a global search of the search space with the aim of identifying the optimal area. The mathematical model of the first phase is described as follows:(10)$$P_{i}=X_{k},i=1,2,\ldots ,N,k=1,2,\ldots ,i-1,i+1,\ldots ,N$$
(11)$$x_{i,j}^{\mathrm{new},P1}=\left\{\begin{array}{c} x_{i,j}+r\left(p_{i,j}-Ix_{i,j}\right),F_{P_{i}}<F_{i},\\ x_{i,j}+r\left(x_{i,j}-p_{i,j}\right),F_{P_{i}}\geq F_{i}, \end{array}\right.$$
(12)$$X_{i}=\left\{\begin{array}{l} X_{i}^{\mathrm{new},P1},F_{i}^{\mathrm{new},P1}<F_{i}\\ X_{i},F_{i}^{\mathrm{new},P1}\geq F_{i} \end{array}\right.$$
where $P_{i}$ is the position of prey for the *i*-th northern goshawk, $F_{P_{i}}$ is its objective function value, *k* is a random natural number in interval [1,N], $X_{i}^{\mathrm{new},P1}$ is the new status for the *i*-th proposed solution, $x_{i,j}^{\mathrm{new},P1}$ is its *j*-th dimension, $F_{i}^{\mathrm{new},P1}$ is its objective function value based on the first phase of NGO, *r* is a random number in interval [0,1], and *I* is a random number that can be 1 or 2. Parameters *r* and *I* are random numbers used to generate random NGO behavior in search and update.

(2)Phase 2: Chase and Escape Operation (Exploitation)

After the northern goshawk attacks the prey, the prey tries to escape. Therefore, in a tail and chase process, the northern goshawk continues to chase the prey. Due to the high speed of northern goshawks, they can chase their prey in almost any situation and eventually hunt. Simulation of this behavior increases the exploitation power of the algorithm to a local search of the search space. In the NGO algorithm, it is assumed that this hunting is closed to an attack position with radius R. The mathematical model of the second phase is as follows:(13)$$x_{i,j}^{\mathrm{new},P2}=x_{i,j}+R(2r-1)x_{i,j}$$
(14)$$R=0.02\left(1-\frac{t}{T}\right)$$
(15)$$X_{i}=\left\{\begin{array}{l} X_{i}^{\mathrm{new},P2},F_{i}^{\mathrm{new},P2}<F_{i}\\ X_{i},F_{i}^{\mathrm{new},P2}\geq F_{i} \end{array}\right.$$
where *t* is the iteration counter, T is the maximum number of iterations, $X_{i}^{\mathrm{new},P2}$ is the new status for *i*-th proposed solution, $x_{i,j}^{\mathrm{new},P2}$ is its *j*-th dimension, and $F_{i}^{\mathrm{new},P2}$ is its objective function value based on the second phase of NGO.

Through the above solving process, NGO demonstrates relatively high convergence accuracy and good stability, yet it still has several limitations [24]. Firstly, during the initialization of the population, more random distributions and uneven initial solutions are generated, which may decrease the diversity of the population, leading to the algorithm’s inability to find the optimal solution. Secondly, during the search, each dimension of the individual eagle decreases, gradually narrowing the search space and increasing the probability of the algorithm falling into local space. Thirdly, northern goshawks chase their escaping prey at a high speed, causing the algorithm to search too fast in the later stages and falling into local optima.

In response to the above-mentioned issues, this paper proposes improvements to the NGO algorithm using cubic chaotic mapping, a hybrid sine–cosine algorithm, and a random differential perturbation strategy, respectively.

(1)Cubic chaotic mapping

Chaotic mapping is utilized to enhance the traversal and uniformity of initial solutions, thus improving the algorithm’s global search capability. Consequently, this paper employs cubic chaotic mapping to initialize the population individuals of the NGO algorithm. Its formula is described as follows:(16)$$x_{n+1}=\rho x_{n}\left(1-{x_{n}}^{2}\right)$$

(2)Hybrid sine–cosine algorithm

The hybrid sine–cosine algorithm utilizes the concept of sine and cosine functions to accomplish global exploration and local exploitation of the search space. Furthermore, the introduction of the cosine factor helps enhance the algorithm’s local exploitation capability, thereby avoiding the risk of falling into local optima. The mathematical formula for the cosine factor and the improved expression for the goshawk position are presented as follows:(17)$$\omega =\cos\left(\pi \frac{t^{3}}{T^{3}}\right)\left(\omega_{\max}-\omega_{\min}\right)+\omega_{\min}$$
(18)$$x_{i,j}^{\mathrm{new},P_{3}}=\left\{\begin{array}{l} (1-\omega )x_{i,j}+\omega \cdot \sin\left(r_{1}\right)\left(r_{2}p_{i,j}-x_{i,j}\right)\\ F_{P_{i}}<F_{i}\\ (1-\omega )x_{i,j}+\omega \cdot \cos\left(r_{1}\right)\left(x_{i,j}-r_{2}p_{i,j}\right)\\ F_{P_{i}}⩾F_{i} \end{array}\right.$$

(3)Random differential perturbation strategy

In the later stages of the algorithm, the northern goshawk tends to confuse local optima with global optima during its pursuit of prey, leading the algorithm to fall into local optima. To overcome this drawback, the random differential perturbation strategy is introduced.
(19)$$X_{t+1}=r\times (X_{best}-X_{i})+r\times (X_{rand}-X_{i})$$

### 2.3. The Weight Allocation Method Based on the MIC

Due to labeled data originating from multiple working conditions, utilizing a small amount of labeled data from a single condition alone for knowledge transfer cannot achieve the desired predictive performance. Therefore, this paper considers knowledge transfer from multiple source conditions. The fermentation process data under a source condition are considered as a subset. The model established on this subset after TCA adaptation is referred to as a sub-model. This paper establishes multiple sub-models and introduces a weighted ensemble strategy based on the MIC to allocate weights more reasonably to each sub-model. The weighted ensemble strategy based on the MIC proposed in this paper mainly consists of two key points. Firstly, it involves utilizing the MIC to compute the centroid of each subset. Secondly, it utilizes the MIC to calculate the correlation between the test sample and the centroid of each subset in order to allocate suitable weight for each sub-model.

The maximum information coefficient (MIC) is mainly used to measure the degree of correlation between two variables, which is an excellent method of data correlation calculation [25].

#### 2.3.1. Determination of the Centroid of Each Subset

Assuming that there are *q* subsets, X=xi;i=1,2,⋯nq represents q subsets, where xi∈ℝd, *d* represents the input dimension, and nq represents the number of samples of each subset. Now, assuming each sample serves as a reference sequence once, with the remaining samples acting as comparison sequences, the MIC is computed using mutual information and grid partitioning, with the formula:(20)Ix0;xi=∫px0,xilog2px0,xipx0pxidx0dxi
where px0,xi represents the joint probability between x0,xi. Considering D=x0(1),xi(1),⋯,x0(d),xi(d) as the comparison sequence, the scatter plot formed by x0,xi in D is partitioned into grids, calculating the probability of each grid, and determining the maximum mutual information value.
(21)φi=maxab<BIx0;xilog2min(a,b)
where *a*, *b* represents the number of grid divisions in the x0,xi direction and B is the upper limit value for grid partitioning. According to Formula (24), the correlation matrix generated by nq samples is as follows:(22)ϕ=1φ12⋯φ1nqφ211⋯φ2nq⋮⋮⋱⋮φnq1φnq2⋯1

The sample with the highest correlation with all comparison sequences is chosen as the initial centroid $Z^{*}$ of the subset. The sample is $x_{k}$, where $k\in \left[1,n_{q}\right]$ is determined according to Formula (25) to calculate the mutual information of each feature variable under the maximum correlated sequence, denoting the mutual information value of the *d*-th feature vector of the *i*-th sample $I\left[x_{k}(d);x_{i}(d)\right]$ as $I_{i}(d)$; then, the correlation coefficient matrix of each indicator can be represented as follows:(23)$$\psi =\left[\begin{array}{cccc} I_{1}(1) & I_{1}(2) & \cdots & I_{1}(d)\\ I_{2}(1) & I_{2}(2) & \cdots & I_{2}(d)\\ \vdots & \vdots & \ddots & \vdots \\ I_{n_{q}}(1) & I_{n_{q}}(2) & \cdots & I_{n_{q}}(d) \end{array}\right]$$

In order to obtain a more objective centroid for the subset, information entropy is introduced to assign corresponding weights to feature variables. In general, there is an inverse relationship between the information entropy of feature variables and the assigned weights; smaller information entropy indicates a greater amount of information carried by the data, hence correspondingly larger weights, and vice versa. According to Formula (26), the proportional representation of feature value weights for the *i*-th sample under the d-th feature vector can be expressed as:(24)$$\rho_{id}=\frac{I_{n_{q}}(d)}{\sum_{i=1}^{n_{q}}I_{n_{q}}(d)}$$

The corresponding entropy value is:(25)$$\left\{\begin{array}{l} E_{d}=-\frac{\sum_{i=1}^{n_{q}}\rho_{id}\log\rho_{id}}{\log d},\rho_{id}\neq 0\\ \underset{\rho_{id}\to 0}{\lim}\rho_{id}\log\rho_{id}=0,\rho_{id}\neq 0 \end{array}\right.$$

Then, the weight of each feature variable is
(26)$$\omega_{d}=\frac{1-E_{d}}{d-\sum_{i=1}^{d}E_{d}}$$

Ultimately, the weighted centroid set $Z_{q}=\omega_{d}Z^{*}$ can be obtained, representing *q* centroids of the *q* subsets.

#### 2.3.2. The Selection and Weighting Allocation Strategy for Sub-Models

Above, the method of obtaining centroids that best represent each subset is introduced and obtains *q* centroids for *q* subsets. Then, by calculating the correlation between the sample to be tested and these centroids, an appropriate weight is assigned to each sub-model.

The *q* sub-models are denoted as $M=\left[M_{1},M_{2},\cdots ,M_{q}\right]$. To improve predictive performance, ensemble models should be constructed by selecting the sub-models corresponding to subsets which exhibit the highest correlation with the test sample. Therefore, by treating the test sample *x** under the target condition as the reference sequence and the *q* centroids $Z_{q}$ as the comparison sequences, the correlation set $O=\left[w_{1},w_{2},\cdots ,w_{q}\right]$ can be obtained by calculating the correlation between the test sample *x** and *q* centroids $Z_{q}$. When $w_{q}\geq \theta ,\theta \in \left[\min\left(w_{q}\right),\max\left(w_{q}\right)\right]$ is determined, the corresponding sub-models are selected for weighted ensemble.

Assuming the selected sub-models are denoted as $M^{\prime }=\left[M_{1},M_{2},\cdots ,M_{\delta }\right]$, δ∈[1,q], and the corresponding correlation set is $O^{\prime }=\left[w_{1},w_{2},\cdots ,w_{\delta }\right],\;\delta \in [1,q]$, then the predictive results of the model based on the MIC-weighted ensemble are:(27)$$y^{*}=\frac{w_{1}}{\sum w_{\delta }}f_{1}\left(x^{*}\right)+\frac{w_{2}}{\sum w_{\delta }}f_{2}\left(x^{*}\right)+\cdots +\frac{w_{\delta }}{\sum w_{\delta }}f_{\delta }\left(x^{*}\right)$$
where, $f_{\delta }\left(x^{*}\right)$ represents the prediction result of the δ-th sub-model.

### 2.4. Data Acquisition and Soft Sensor Modeling

#### 2.4.1. The Fermentation Process and Data Acquisition of Pichia Pastoris

To validate the effectiveness of the proposed soft sensor method, *Pichia pastoris* strains GS115 and MutsHis+ were selected as the research subjects. The fermentation setup utilized the RTY-C-100L fermenter. Following an in-depth analysis of absolute correlation within the *Pichia pastoris* fermentation process, the input variables for the soft sensing model were meticulously chosen. These encompassed parameters such as stirring speed (*v*), temperature (*T*), airflow rate (*q*), pH of the culture medium (*Ph*), dissolved oxygen concentration (*Do*), and fermenter pressure (*P*). Concurrently, product concentration and cell concentration were designated as the output variables, as depicted in Figure 1. The precise procedures involved in acquiring the experimental data for *Pichia pastoris* fermentation are outlined below.

Step 1. In adherence to the fermentation process specifications for *Pichia pastoris*, the fermentation system underwent sterilization and inoculation steps. The culture medium was sterilized at 130 °C for 30 min, followed by inoculation of strains at 30 °C using a flame. Initial fermentation conditions were set as follows: tank pressure maintained between 0.02 and 0.05 MPa, pH at 5.0, temperature at 28 °C, stirring speed ranging from 300 to 400 rpm, and airflow rate regulated between 150 and 300 L/min.

Step 2. Using correlation analysis, variables including stirring speed (*v*), temperature (*T*), airflow rate (*q*), pH (*Ph*), dissolved oxygen (*Do*), and fermenter pressure (*P*) were selected. These variables were transmitted to the database via a distributed control system, with sampling conducted every half-hour.

Step 3. Owing to the 90 h fermentation cycle of *Pichia pastoris*, 180 data points were collected for each batch of the fermentation process. Fermentation process data from different batches were collected to represent the fermentation process under different working conditions.

#### 2.4.2. Soft Sensor Modeling Based on MIC-TCA-INGO-LSSVM

In this paper, a novel soft sensor model based on MIC-TCA-INGO-LSSVM is proposed to address the issue of soft sensor model failure caused by inconsistent data distribution during the fermentation process of *Pichia pastoris* under different working conditions. The proposed strategy integrates the principles of transfer learning and ensemble learning. It utilizes TCA to adapt the marginal probability distributions of fermentation process data from both the source and target working conditions. Additionally, a weighted ensemble scheme based on the MIC is introduced to transfer knowledge from multiple source conditions, thereby enhancing the model’s generalization ability and predictive accuracy under the target working condition. Furthermore, an INGO algorithm is proposed to enhance the optimization capability of the NGO algorithm for optimizing the LSSVM model. Algorithm 1 gives a description of the proposed method MIC-TCA-INGO-LSSVM.

*q* subsets with known labels are selected as the source domain modeling data, and fermentation process data under the target working condition are chosen as the target domain test data with unknown labels. The steps for building a soft sensor model based on MIC-TCA-INGO-LSSVM, with the source domain and target domain data denoted as $D_{s}=\left\{X_{\mathrm{si}},Y_{\mathrm{si}};i=1,2\ldots q\right\}$ and $D_{t}=\left\{X_{t}\right\}$ respectively, are as follows:

Step 1. Utilize TCA to obtain the optimal feature mapping matrix, adapting the marginal distributions of the source and target data: $\left(X_{\mathrm{si}}^{\prime },X_{t}^{\prime }\right)=TCA\left(X_{\mathrm{si}},X_{t}\right)$, i=1,2…q. Where $X_{\mathrm{si}}^{\prime }$ represents the *i*-th new source domain subset and $X_{t}^{\prime }$ represents the new target domain data after being adapted by TCA, respectively.

Step 3. Combining with INGO-optimized LSSVM, establish a sub-model for each new subset along with their corresponding labels $\left(X_{s1}^{\prime },Y_{\mathrm{sl}}\right)$, $\left(X_{s2}^{\prime },Y_{s2}\right)$… $\left(X_{\mathrm{sq}}^{\prime },Y_{\mathrm{sq}}\right)$.

Step 4. Compute the centroid of each new source domain subset using the MIC-based method introduced in Section 2.3.1. Finally, obtain *q* centroids, denoted as $Z_{1},Z_{2}\ldots Z_{q}$.

Step 5. Calculate the maximum information coefficient between the test sample $x_{t}^{\prime }$ under the target working condition and the centroids of each subset, forming a set of correlation degrees, denoted as $O=\left[w_{1},w_{2},\cdots ,w_{q}\right]$. Select sub-models with correlation coefficients greater than θ. Use the method described in Section 2.3.2 to assign corresponding weights to each sub-model. Input the test sample $x_{t}^{\prime }$ into the ensemble transfer soft sensor model MIC-TCA-INGO-LSSVM to obtain the cell concentration and product concentration results $Y_{t}$.
**Algorithm 1**: Soft Sensor Modeling Method MIC-TCA-INGO-LSSVM.**Input:**S: Source domain data with known labels.T: Target domain data with unknown labels.**Output:**Predicted cell concentration and product concentration for the target working condition.Steps:**1. Data Preparation**Collect fermentation process data under multiple-source working conditions and the target working condition.Select key biological parameters as input variables (e.g., stirring speed, temperature, and airflow rate).**2. Transfer Component Analysis (TCA)**Apply TCA to adapt the marginal distributions of the data from source to target working conditions.Obtain a feature mapping matrix that minimizes the difference in data distribution between source and target.**3. Model Construction with LSSVM Optimized by INGO**For each adapted source data subset, construct a least squares support vector machine (LSSVM) model.Use the improved northern goshawk optimization (INGO) algorithm to optimize the parameters of each LSSVM model.**4. Weight Allocation Based on the MIC**Calculate the maximum information coefficient (MIC) to determine the centroid for each source data subset.For a given test sample in the target domain, calculate the MIC between the test sample and each subset centroid.Assign weights to each sub-model based on the MIC values, facilitating a weighted ensemble.**5. Ensemble Prediction**Combine the predictions from the weighted sub-models to estimate the cell and product concentrations in the target working condition.**6. Performance Evaluation**Evaluate the model using metrics like Root Mean Square Error (*RMSE*), Correlation Coefficient (*R*^2^), and Mean Absolute Error (*MAE*).**End**

In order to display the soft sensor model more intuitively, Figure 2 shows the framework of the model.

To quantify the predictive capabilities of various models, this study selects three metrics for comparison: root mean square error (*RMSE*), correlation coefficient (*R*^2^), and mean absolute error (*MAE*). The computational formulas for these metrics are provided as follows:(28)$$RMSE=\sqrt{\frac{1}{n}\sum_{i=1}^{n}{\left(y_{t}^{\left(i\right)}-y_{\mathrm{pre}}^{\left(i\right)}\right)}^{2}}$$
(29)$$R^{2}=1-\frac{\sum_{i=1}^{n}{\left(y_{\mathrm{pre}}^{\left(i\right)}-y_{\mathrm{real}}^{\left(i\right)}\right)}^{2}}{\sum_{i=1}^{n}{\left(y_{{\;}_{\mathrm{real}}}^{\left(i\right)}-{\hat{y}}_{\mathrm{real}}\right)}^{2}}$$
(30)$$MAE=\frac{1}{n}\sum_{i=1}^{n}\left|y_{t}^{\left(i\right)}-y_{\mathrm{pre}}^{\left(i\right)}\right|$$

## 3. Results

To validate the effectiveness of the proposed soft sensor model, MIC-TCA-INGO-LSSVM, based on TL and EL, it is applied to predict the cell concentration and product concentration under the target working condition of *Pichia pastoris*.

Firstly, to demonstrate the performance of the INGO algorithm proposed in this paper, simulation experiments are conducted under the same working condition. Three models, namely, LSSVM, NGO-LSSVM, and INGO-LSSVM, are employed to predict the cell concentration and product concentration. The *RMSE*, *MAE*, and *R*^2^ results obtained for predicting cell concentration and product concentration by the three models are presented in Table 1. Figure 3 illustrates the predicted results of cell concentration and the fitness changes of the optimization algorithms, while Figure 4 illustrates the predicted results of product concentration and the fitness changes of the optimization algorithms, where the fitness functions of the NGO and INGO algorithms are defined as the *RMSE* between the predicted results and the actual values. From Figure 3 and Figure 4 and Table 1, it can be seen that compared to NGO, INGO demonstrates faster convergence speed and is capable of achieving smaller prediction errors, thereby improving the predictive precision of the LSSVM. It can be proven that the northern goshawk algorithm is improved by using cubic chaotic mapping and a hybrid sine–cosine algorithm, and a random differential perturbation strategy is feasible for training and optimizing parameters of the LSSVM models, and the effect is obvious: it can significantly reduce the prediction error of the LSSVM. Comparative experiments indicate that under the same working condition, when the data distribution remains consistent, all three traditional soft sensor models yield relatively good predictive results.

Subsequently, simulation comparisons are conducted under different working conditions. Initially, fermentation process data from a single-source working condition are selected as modeling data. NGO-LSSVM and INGO-LSSVM are applied once again and compared with TCA-INGO-LSSVM for predicting the cell concentration and product concentration under the target working condition. Table 2 presents the predictive performance of the three models when the working conditions varied. Figure 5 and Figure 6, respectively, depict the predicted results of cell concentration and product concentration under the target working condition. By comparing the predictive performance of NGO-LSSVM and INGO-LSSVM models in Table 1 and Table 2, it can be observed that the predictive performance of traditional soft sensor models decreases when working conditions change. However, TCA narrows the distribution difference between the fermentation data under the source working condition and fermentation data under the target working condition by adapting the marginal probability distribution, thereby mitigating the difference between them. Therefore, as shown in Table 2, TCA improves the model’s ability to adapt to changes in working conditions to some extent, thereby reducing the prediction error of the model under the target condition.

Nevertheless, solely conducting knowledge transfer from a single-source working condition restricts the model’s generalization and predictive capabilities under the target working condition. Therefore, the proposed soft sensor model, MIC-TCA-INGO-LSSVM, based on transfer learning and ensemble learning, is applied to predict cell concentration and product concentration under the target working condition. Figure 7 and Figure 8 respectively illustrate the predicted results of cell concentration and product concentration. Where, subplot ‘a’ represents the equally weighted ensemble approach of multiple sub-models, while subplot ‘b’ represents the MIC-based weighted ensemble method proposed in this study. Table 3 compares the predictive performance of the two models. By comparing Table 2 and Table 3, it is evident that transferring knowledge from multiple working conditions yields better predictive outcomes compared to transferring from a single working condition. Particularly, the proposed MIC-based weighted ensemble method effectively allocates suitable weights to each sub-model, resulting in smaller prediction errors compared to the equally weighted ensemble approach.

## 4. Discussion

This paper proposes a novel soft sensor modeling method for the fermentation process of *Pichia pastoris*, named MIC-TCA-INGO-LSSVM. Simulation results indicate that this method significantly reduces the prediction error of traditional soft sensor models under varying working conditions, aligning with the expectations outlined in the paper.

Under stable working conditions, traditional soft sensor models demonstrate robust predictive performance [26,27]. However, as industrial demands evolve, the working conditions for the fermentation process of *Pichia pastoris* are subject to change, leading to a noticeable increase in the predictive error of conventional models. Consequently, researchers have proposed soft sensor modeling methods based on transfer learning, which have proven effective in mitigating model errors amidst changing conditions [9,10,11,28,29]. Further, this paper hypothesizes that with limited labeled samples available from a single-source working condition, knowledge transfer from just one source is insufficient to effectively enhance predictive performance under target conditions. It is experimentally demonstrated that knowledge transfer from multiple-source working conditions and model ensemble weighting more reasonably enhance the model’s generalization capability and predictive accuracy under unknown working conditions. This suggests that when there are few labeled samples in a single working condition, the multi-condition transfer soft sensor modeling method proposed in this paper is more effective compared to existing methods, serving as a valuable reference.

However, the fermentation process of *Pichia pastoris* exhibits multi-stage characteristics, and the global single soft sensor model proposed in this paper cannot fully describe the dynamic characteristics of the entire fermentation process. Therefore, theoretically, combining the proposed modeling method with local modeling strategies in future research could enhance the predictive capability of the model. Moreover, the transfer component analysis used in this paper is a transfer learning algorithm that can be directly applied to regression problems but shows suboptimal transfer effectiveness. Thus, future research could consider improving and applying more efficient transfer learning algorithms in soft sensor modeling.

## 5. Conclusions

Given the varied data distribution under different working conditions of the fermentation process of *Pichia pastoris*, which results in a decline in the performance of traditional soft sensor models, this study proposes a soft sensor modeling method based on TCA and EL. Firstly, TCA is employed to adapt the marginal probability distributions of modeling data under the source working condition and testing data under the target working condition. Then, a regression model is established using the LSSVM on the new source working condition data after adapting, with an INGO algorithm proposed for optimizing the parameters of the LSSVM. Secondly, considering that knowledge transfer from a single-source working condition would limit the model’s generalization and predictive performance under the target working condition, multiple sub-models are established on multiple-source working condition datasets and a weighting allocation method based on the MIC is proposed to assign suitable weights to each sub-model. The proposed soft sensor model, MIC-TCA-INGO-LSSVM, is then applied to predict cell concentration and product concentration under the target working condition. Results demonstrate: (1) Compared to NGO, INGO demonstrates a faster convergence speed and is capable of achieving smaller prediction errors, thereby improving the predictive precision of the LSSVM. (2) The transfer soft sensor model integrated with multiple sub-models significantly enhances the model’s adaptability and predictive capability under different working conditions. Moreover, compared to the average weighting allocation method, the MIC-based weighting allocation method achieves smaller prediction errors. In conclusion, the soft sensor model MIC-TCA-INGO-LSSVM proposed in this paper can be applied for real-time and accurate detection of cell concentration and product concentration in the fermentation processes of *Pichia pastoris* under different working conditions.

## Figures and Tables

**Figure 1 sensors-24-03017-f001:**
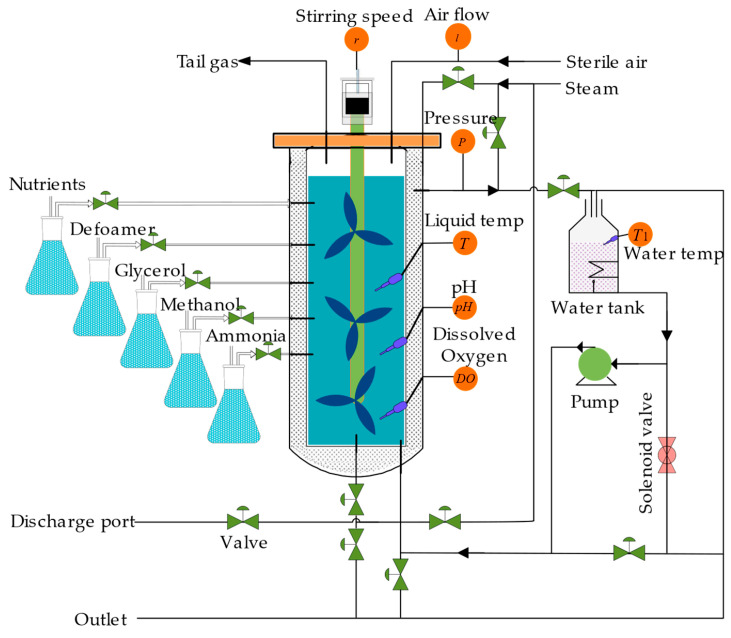
*Pichia pastoris* fermentation system.

**Figure 2 sensors-24-03017-f002:**
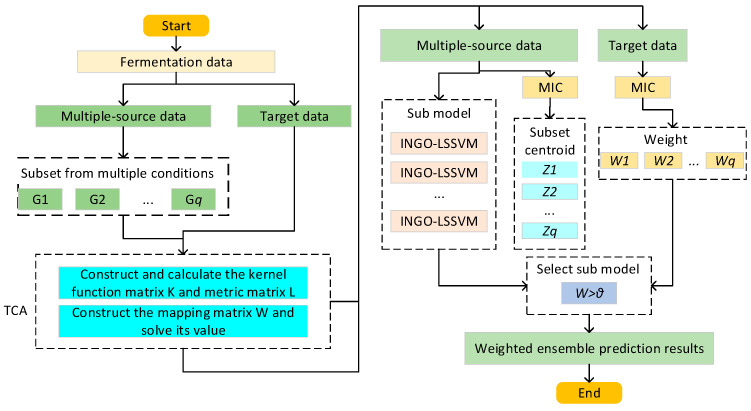
The framework of the MIC-TCA-INGO-LSSVM.

**Figure 3 sensors-24-03017-f003:**
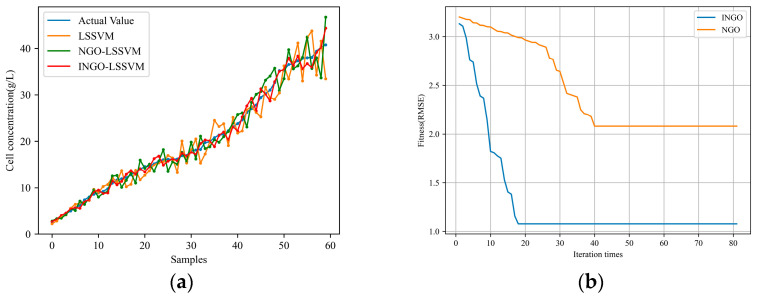
Prediction results of cell concentration under the same working condition: (**a**) prediction results; (**b**) the fitness curve of the INGO and NGO algorithm.

**Figure 4 sensors-24-03017-f004:**
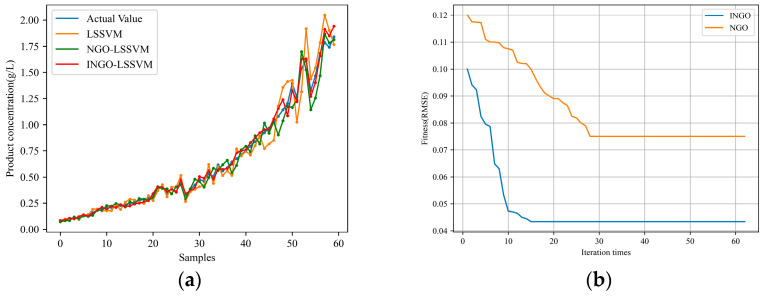
Prediction results of product concentration under the same working condition: (**a**) prediction results; (**b**) the fitness curve of the INGO and NGO algorithm.

**Figure 5 sensors-24-03017-f005:**
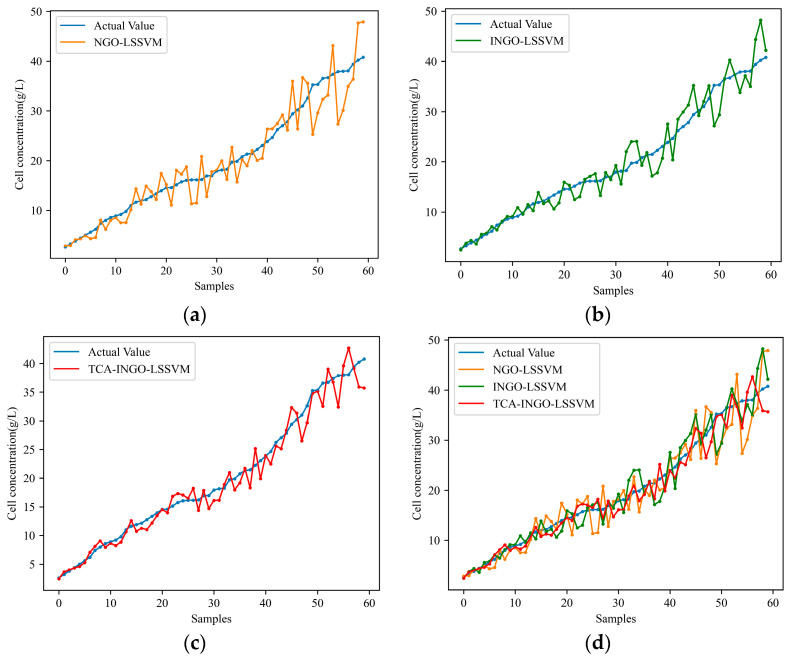
Prediction results of cell concentration with changing working conditions: (**a**) NGO-LSSVM; (**b**) INGO-LSSVM; (**c**) TCA-INGO-LSSVM; and (**d**) combination of the three models.

**Figure 6 sensors-24-03017-f006:**
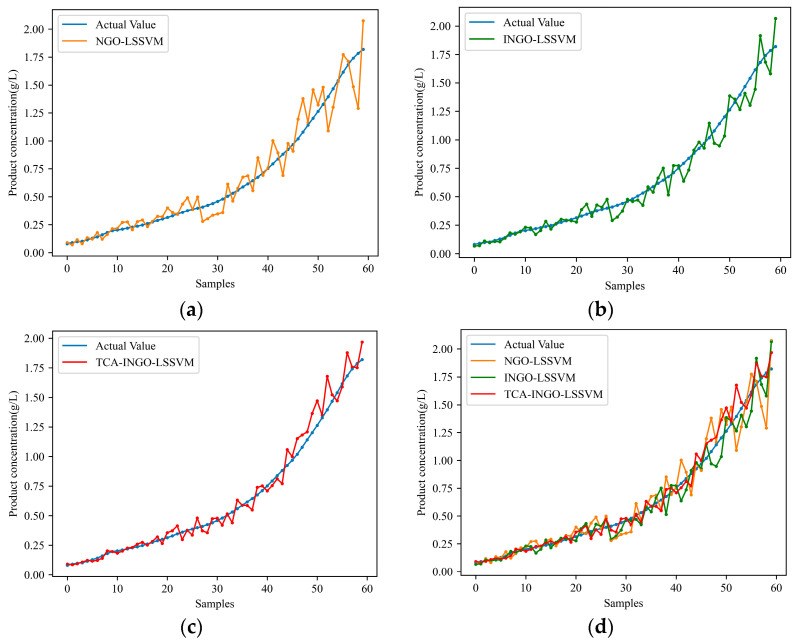
Prediction results of product concentration with changing working conditions: (**a**) NGO-LSSVM; (**b**) INGO-LSSVM; (**c**) TCA-INGO-LSSVM; and(**d**) combination of the three models.

**Figure 7 sensors-24-03017-f007:**
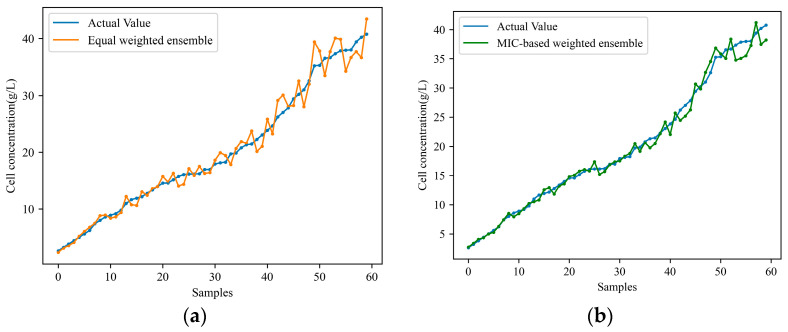
Ensemble multiple sub-models for predicting cell concentration in the target condition: (**a**) equally weighted ensemble; (**b**) MIC-based weighted ensemble.

**Figure 8 sensors-24-03017-f008:**
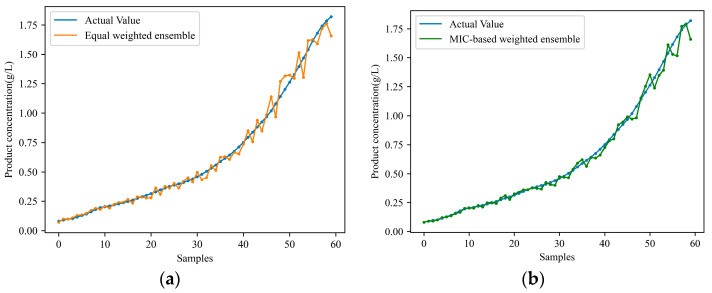
Ensemble multiple sub-models for predicting product concentration in the target condition: (**a**) equally weighted ensemble; (**b**) MIC-based weighted ensemble.

**Table 1 sensors-24-03017-t001:** The predictive performance of different models for predicting cell and product concentration under the same working condition.

		*RMSE*	*MAE*	*R* ^2^
Cell concentration	LSSVM	2.45	1.84	0.950
	NGO-LSSVM	2.05	1.54	0.965
	INGO-LSSVM	1.08	0.79	0.991
Product Concentration	LSSVM	0.099	0.068	0.961
	NGO-LSSVM	0.076	0.051	0.977
	INGO-LSSVM	0.044	0.032	0.992

**Table 2 sensors-24-03017-t002:** The predictive performance of different models for predicting cell and product concentration with changing working conditions.

		*RMSE*	*MAE*	*R* ^2^
Cell concentration	NGO-LSSVM	3.71	2.83	0.883
	INGO-LSSVM	2.85	2.14	0.921
	TCA-INGO-LSSVM	2.02	1.52	0.967
Product Concentration	NGO-LSSVM	0.146	0.100	0.915
	INGO-LSSVM	0.091	0.061	0.966
	TCA-INGO-LSSVM	0.067	0.047	0.982

**Table 3 sensors-24-03017-t003:** The performance of the soft sensor method, integrating multiple sub-models, in predicting cell and product concentration under the target working condition.

		*RMSE*	*MAE*	*R* ^2^
Cell concentration	equal ensemble	1.67	1.31	0.977
	weighted ensemble	1.18	0.88	0.989
Product concentration	equal ensemble	0.058	0.045	0.987
	weighted ensemble	0.046	0.034	0.991

## Data Availability

The dataset in this article is unavailable because it involves privacy.

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
