# Peer review of "Modeling and Optimization of an Enhanced Soft Sensor for the Fermentation Process of Pichia pastoris"

_sensors, 2024, doi:10.3390/s24103017_

Round 1

Reviewer 1 Report

Comments and Suggestions for Authors

Concerning the work submitted by Wang et al. entitled “ Modeling and Optimization of an Enhanced Soft Sensor for the Fermentation Process of Pichia pastoris Based on TCA and MIC” I read it with interesting. The work is well written, has some novelty and the data is acceptable for publication. I therefore recommend for publication in recent form.

Author Response

Thank you very much for your thorough review and positive feedback on our manuscript. We greatly appreciate your time and effort in evaluating our work.Your endorsement is highly encouraging, and we are grateful for your confidence in the quality of our research.We will proceed with the publication process as per your suggestion and look forward to sharing our findings with the scientific community. Once again, thank you for your valuable feedback and support.

Reviewer 2 Report

Comments and Suggestions for Authors

This is a nice paper, with good work and well written. I would accept it with minor revisions. HOWEVER, surprisingly, there is no discussion section or any kind of discussion of their main findings. The authors should discuss the results and how they can be interpreted in terms of previous studies and the working hypotheses. The results and their implications should be discussed in the broadest possible context, and limitations of the work should be highlighted. Future research directions can also be mentioned. 

Title: I am not a big fan of abbreviations in titles... maybe the authors could remove it and leave in the keywords.

Keywords: avoid repeating the same words that are already in the title.

Pichia pastoris should always be in italic. Do it throughout the text.

Abstract: It is well written. However, the authors spend a lot of words introducing and explaining methodologies, as a consequence of that, little is described regarding their main findings and conclusions. Therefore, I suggest the authors to restructure it. 

Introduction: it is clear and well written. I suggest the authors to make it shorter and more concise. For instance, paragraph 2 can be shortened by half. The same for paragraph in lines 95 to 134.

Materials and methods: lines 159 to 161-> any reference for this sentence?

Data Acquisition and Soft Sensor Modeling -> maybe it could be merged with the materials and methods?

Results: well written

Conclusions: well written

Author Response

Thank you for giving us the opportunity to submit a revised draft of the manuscript “Modeling and Optimization of an Enhanced Soft Sensor for the Fermentation Process of Pichia pastoris Based on TCA and MIC” (sensors-2961200) for publication in the Sensors, We appreciate the time and effort that you dedicated to providing feedback on our manuscript and are grateful for the insightful comments on and valuable improvements to our paper. And now we response the reviewer comments with a point by point.

Reviewer 3 Report

Comments and Suggestions for Authors

This paper proposes a soft sensor modeling method named MIC-TCA-INGO-LSSVM to address performance degradation in soft sensor models caused by this scenario. Generally, the paper is of scientific sounds. Before possible publication, the following concerns must be addressed.

1. The reviewer suggest the authors rewrite the Abstract. Currently, the length of Abstract is too long. The authors can remove the detailed description of the method and point out the main idea of the proposed method.

2. The authors said that the proposed method enhanced the predictive accuracy significantly in Abstract. So, it is recommended that the improvements of predictive accuracy be placed here to highlight the significant advantages of the method. 

3. The expression of the third paragraph in Introduction is problematic. For example, there are two consecutive "however" statements. The reviewer suggests the authors reorganize this paragraph. 

4. Although the authors have introduced many TCA work.  It is still need to enhance the motivation of the use of TCA in the proposed method.

5. The introduction should end with a chapter organization.

6. The font size of formulas should be consistent with the text.

7. Generally, Least Squares Support Vector Machine is known to most readers. So, Section 2.2.1 can be compressed, that is, only the main idea of the algorithm is written.

8. The layout of the paper should be improved. For example, the last line of formula (9) is not fully displayed.

9. It is suggested that an algorithm be written to describe the proposed method.

10. The author should clarify which ensemble learning method is used in this paper.

11. The running time of the proposed method is missing. Alternatively, the authors can give the time complexity analysis.

12. In addition to summarizing the important findings of this paper, the conclusion should also point out the shortcomings of this study.

Author Response

(The authors gave the same response as above.)

Round 2

Reviewer 2 Report

Comments and Suggestions for Authors

The authors only managed to address simple comments. And for most of the major concerns they did not address properly.

I had made it clear that the Discussion section was lacking. Although the authors add a Discussion section, it is basically one paragraph in which the authors fail to compare their findings with other studies. 

Then I decided to look for other works from these authors and found a very similar study published in this same journal last year. The authors did not even use their previous study to discuss their findings. And, surprisingly, in their previous study they also failed to have a discussion section in which they properly compare their findings and future perspectives with the literature. 

Having a proper discussion using the recent literature in the topic to contextualize your work and main findings is one of the BASIC things of a scientific article (tip: https://www.ncbi.nlm.nih.gov/pmc/articles/PMC3474301/). I really considered to reject this article because the authors did not address this basic comment. But, if they make a major revision and write a proper Discussion section, I would reconsider. But, if they fail again to do so, I would have to suggest a rejection. 

Comments on the Quality of English Language

Should have a proofread. 

Author Response

Thank you for your continued review and valuable feedback. We apologize for the oversight in adequately addressing the concerns raised in the previous review. We acknowledge the importance of a comprehensive Discussion section and understand the necessity of comparing our findings with relevant literature to contextualize our work effectively. We will carefully revise the Discussion section to ensure a thorough comparison with previous studies and provide insights into the implications and future directions of our research. We are committed to addressing all major concerns raised by the reviewer and ensuring the quality of our manuscript. Your guidance is greatly appreciated, and we will strive to meet your expectations in this revision.

I have revised the content of the Discussion section in the original manuscript and have stored the modified content in the "Reviewer2.PDF" file.

Once again, I would like to express my gratitude for your guidance in my academic writing. In future academic writing endeavors, I will pay close attention to the composition of the Discussion section and strive to enhance the completeness of my papers.

Reviewer 3 Report

Comments and Suggestions for Authors

The authors have addressed all concerns. I have no further comments.

Author Response

Thank you for your continued review of our manuscript. We appreciate your thoroughness and are pleased to hear that our revisions have satisfactorily addressed all concerns. Your feedback has been invaluable in improving the quality of our work. We look forward to the next steps in the publication process.

Round 3

Reviewer 2 Report

Comments and Suggestions for Authors

Dear authors and editors,

I've been thinking and concluded that in a first moment, a paper without Discussion should be rejected or suggested to be resubmitted. I should have not even be considered for a peer reviewing process. This is a minimum requirement. 

I insist on that, as if this journal wants to keep its high standard, they editorial board and reviewers should be more critical not only about the research content but also regarding the manuscript structure.

This is the third version of this manuscript. The authors are clearly not putting enough effort on writing a proper Discussion. Therefore, I am also not putting more efforts on keep reviewing this manuscript and suggest it to be rejected. 

The content is more important than the length. Putting some words together just to 'meet' the reviewers' requirements is not enough to get a good Discussion of your research.

Comments on the Quality of English Language

Can be improved

Author Response

Dear reviewer,

Firstly, thank you very much for your careful and detailed review of this manuscript. In accordance with your suggestion, I have once again delved into and understood the writing requirements and tips for the "discussion" section in the paper.

At present, I have rewritten the "discussion" section and provided it in the attachment. In the rewritten "discussion", I not only summarized the conclusions of the article but also compared it with previous studies, and pointed out some limitations of the article and the future research directions.

Once again, thank you for your valuable time and patient review.

Best regards,
--
Ameng Yu
